# Slipstreaming Mother Machine: A Microfluidic Device for Single-Cell Dynamic Imaging of Yeast

**DOI:** 10.3390/mi12010004

**Published:** 2020-12-22

**Authors:** David C. Durán, César A. Hernández, Elizabeth Suesca, Rubén Acevedo, Ivón M. Acosta, Diana A. Forero, Francisco E. Rozo, Juan M. Pedraza

**Affiliations:** 1Laboratorio de Biofísica, Departamento de Física, Universidad de los Andes, Bogotá 111711, Colombia; e.suesca87@uniandes.edu.co (E.S.); rl.acevedo10@uniandes.edu.co (R.A.); imacostah@correo.udistrital.edu.co (I.M.A.); daforerog@correo.udistrital.edu.co (D.A.F.); ferozob@correo.udistrital.edu.co (F.E.R.); 2Centro de Microelectrónica, Departamento de Ingeniería Eléctrica y Electrónica, Universidad de los Andes CMUA, Bogotá 111711, Colombia; ca.hernandez11@uniandes.edu.co; 3Proyecto Curricular Licenciatura en Física, Facultad de Ciencias y Educación, Universidad Distrital Francisco José de Caldas, Bogotá 110311, Colombia

**Keywords:** microfluidics, replicative aging, mother machine, *Saccharomyces cerevisiae*

## Abstract

The yeast *Saccharomyces cerevisiae* is one of the most basic model organisms for studies of aging and other phenomena such as division strategies. These organisms have been typically studied with the use of microfluidic devices to keep cells trapped while under a flow of fresh media. However, all of the existing devices trap cells mechanically, subjecting them to pressures that may affect cell physiology. There is evidence mechanical pressure affects growth rate and the movement of intracellular components, so it is quite possible that it affects other physiological aspects such as aging. To allow studies with the lowest influence of mechanical pressure, we designed and fabricated a device that takes advantage of the slipstreaming effect. In slipstreaming, moving fluids that encounter a barrier flow around it forming a pressure gradient behind it. We trap mother cells in this region and force daughter cells to be in the negative pressure gradient region so that they are taken away by the flow. Additionally, this device can be fabricated using low resolution lithography techniques, which makes it less expensive than devices that require photolithography masks with resolution under 5 µm. With this device, it is possible to measure some of the most interesting aspects of yeast dynamics such as growth rates and Replicative Life Span. This device should allow future studies to eliminate pressure bias as well as extending the range of labs that can do these types of measurements.

## 1. Introduction

The yeast *Saccharomyces cerevisiae* has been commonly used as a model organism for aging studies in eukaryotic cells [1,2,3,4,5]. A common measure used in aging studies is the Replicative Life Span (RLS), which is the measure of how long a cell lives while still having the ability to replicate itself [1]. Traditionally, these measurements were done by using the microdissection technique [6,7,8,9,10,11]. In this technique, cells are grown in agar pads and, whenever they replicate, daughter cells are removed with the help of micromanipulators. This technique has several drawbacks: First, it is laborious and time consuming. Second, experiments must be either performed 24/7 or stopped at night while refrigerating the cells under observation, which may bias the interpretation of the results since temperature fluctuations might affect aging. Third, cells may be affected physically by the use of the micromanipulators.

To address these problems, many novel microfluidic devices have been designed for conducting RLS measurements [12,13,14,15,16,17,18,19,20,21,22]. These devices, commonly referred to as Mother Machines, work by trapping cell in a way that they have access to fresh media but are not moving with the flow. Additionally, they are designed in such a way that as the cell replicates the daughter cells are taken away by the flow. Such microfluidic devices work better for RLS measurements than micromanipulators but have two major drawbacks: First, the trapping method subjects cells to mechanical pressure, which could bias the results, as we do not know how pressure affects the aging process in unicellular organisms. It has been demonstrated that mechanical pressure affects physiological processes such as growth rate [23,24] and diffusion rates of intracellular components [25,26]. Second, all the published devices require microfabrication techniques with resolution under 5 µm. This means that custom masks are required, which makes the fabrication process expensive, and thus out of reach for some labs.

To address these issues, we present a microfluidic device that traps cells without subjecting them to high mechanical pressures. To accomplish low-pressure trapping, we take advantage of the slipstreaming effect. This happens when a fluid flows around a barrier and a pressure gradient is created in the back of the barrier, generating a force that points against the direction of the flow. The device is designed in such a way that a gradient zone of the dimensions of a yeast cell is created, allowing the mother cells to be trapped while daughter cells are born outside this area. This means that the mother cells can be kept trapped indefinitely while the daughter cells are rapidly taken away by the flow. This device thus allows measurements of populations of cells at the single-cell level over long periods. Some examples are RLS measurements, distributions of replication times, division strategies, etc. Our device has the added advantage of being fabricated with low-cost techniques since we do not use custom masks but a UV light projector (SF-100 Micropatterning) with 5-µm resolution instead.

In this paper, we present the device and the protocol for its use, as well as a sketch of the optimization used to design it. We present basic gene expression measurements that allow us to conclude that, in the pressure ranges estimated to be used in existing devices, the gene expression pattern is likely to be altered. Finally, we show an example of its use in obtaining the distribution of replication times for a population and an RLS curve and comparing it with the results from previous devices.

## 2. Materials and Methods

### 2.1. Master Design and Fabrication

We used an SF100 Micropatterning UV projector (ScoTech, Renfrewshire, Scotland, UK) instead of a high-resolution printed mask. This projector requires a virtual mask in bmp (bitmap) format, so we designed it pixel by pixel using Illustrator and Paint. One pixel in the design is equivalent to 5 µm by 5 µm in the physical chip. In a cleanroom, we spun SC 1827 Photoresist at 2000 RPM for 1 min and then soft baked it at 115 °C for 50 s. We then exposed it for 50 s to UV light using the projector and developed the master for 1 min before hard baking it at 120 °C for 1 min.

### 2.2. Polydimethylsiloxane (PDMS) Chips Fabrication

Curing agent was mixed with pre-polymer for PDMS in a proportion of 1:10 and cured in an oven at 65 °C. The PDMS was then peeled off the master, and inlet/outlet channels were punched with a 0.75-mm biopsy punch. Chips were then cut and sonicated in isopropanol for 30–40 min, and then blown, dried and heated at 65 °C for 4 h. Glass coverslips were cleaned in 1 M KOH for 30 min, and then sonicated in milliQ water for 10 min. After that, the coverslips were blow-dried and heated at 65 °C for 10 min. The PDMS chips were cleaned with magic tape and treated with oxygen plasma for 1 min for bonding and hot bonded for 1 min at 150 °C.

### 2.3. Chip Dimensions

The device fits on a 22 mm × 22 mm coverslip. Each channel contains an array of 255 trapping units, with 140 μm between units in the direction of the flow and 115 μm between lines of units. Each unit consists of three PDMS pillars of 5-μm height that extend from the ceiling to the glass floor.

### 2.4. Media Preparation

We grew the cells in either Synthetic Dropout (SD) or Synthetic Complete (SC) medium. Every liter of SD medium contained 6700 mg of YNB (Yeast Nitrogen Base), 750 mg of DO supplement, 20 g of D-glucose, 20 mg of uracil and 20 mg of histidine. Every liter of SC medium contained 6700 mg of YNB, 2000 mg of DO supplement, 20 g of D-glucose, 80 mg of D-L tryptophan, 80 mg of D histidine and 80 mg of D-uracil.

### 2.5. Cell Growth and Insertion in Device

We recovered cells from −80 °C storage and grew them in SD Medium (2% Glucose) for 24 h. We diluted 20 μL in 5 mL fresh medium and grew the cells at 30 °C and 245 RPM for 12 h to get them back to an exponential phase (0.05–0.1 OD600). We then took media with these cells and centrifuged it at 5000 RPM for 5 min and then resuspended in 1/5 of the original volume. The cells were then manually inserted into the chip in the direction opposite to the normal flow before turning on the syringe pump and starting the experiment.

### 2.6. Data Acquisition and Analysis

Growth experiments were performed on a Zeiss inverted microscope controlled with Micromanager. We used 40× (air), 60× (air) and 100× (immersion) objectives. Images were acquired using a Photometrics CoolSnap Camera EZ CCD Camera with 1392 × 1040. Media was delivered using a syringe pump (Harvard Apparatus) at a rate of 3.6 mL/h. Expression experiments were performed on a Nikon Ti Eclipse inverted microscope with a 100× (immersion) objective.

### 2.7. Growth Conditions for Measuring Genetic Expression under Pressure

Cells were grown in SD medium supplemented with amino acids (uracil and histidine) and glucose. Cells were grown overnight (24 h) on a plate. We took 20 μL from the overnight culture and diluted it in 10 mL fresh media, and then let it grow for 12 h.

## 3. Results

### 3.1. Fabrication of the Microfluidic Device

We designed and fabricated a novel microfluidic platform (Figure 1A–H) that can be used to measure single cell dynamics in a population of yeast cells. A silicon wafer contains masters for nine devices (Figure 1A). The microfluidic device at the core of the platform has three separate channels that permit the running of experiments with different media or strains independently (Figure 1B). Each channel contains an array of 255 trapping units, with 140 μm between units in the direction of the flow and 115 μm between lines of units (Figure 1C). Each unit consists of three PDMS pillars of 5-μm height that extend from the ceiling to the glass floor (Figure 1D). Cells are trapped to the right of the rightmost pillar while the other two pillars are used for flow focusing (Figure 1D). The z-dimension was chosen such that only one cell fits the channel but is not pressed by the roof of the channel. The device fits in a 22 mm × 22 mm coverslip (Figure 1E). Figure 1F,G shows zoom-ins of the channel and traps. The horizontal dimensions of the device, including unit separation, were determined after several in silico optimization rounds.

The system setup is sketched in Figure 1H. First, the chip is set on the (inverted) microscope and fresh media is loaded into a syringe pump, which is connected to a bubble trap which in turn is connected to the chip inlet. The outlet is connected to a three-way valve, with exits connecting to the waste beaker and a tube used for initial loading of the cells. The cells are then loaded manually through that tube, making sure they do not reach the bubble trap. The three-way valve is then set to the waste beaker tube and flow from the syringe is started. The operation of the trap during this process is sketched in Figure 1I–M.

### 3.2. Optimization of Experimental Procedure

We optimized the experimental procedure for loading and maintenance of the mother cells. To do this, we examined various options for the Optical Density at 600 nm (OD600 or simply OD) at which cells were inserted, the loading procedure and the flow rate for operation. We grew cells to an OD corresponding to peak exponential growth (0.1 OD600), after which we tried two different strategies: inserting cells directly into the chip and concentration by centrifugation before insertion. This was achieved by re-suspending cells in different fractions of the original volume. Concentrating the cells 5× gave the best results in number of trapped cells for various values of OD. Loading the cells in the same direction of the maintenance flow resulted in low trapping rates because the cells mostly flowed around the traps when flow was started, so we settled on loading the cells in the opposite direction before starting the syringe pump. This resulted in a trapping rate of approximately 70%, or about 180 cells per channel. The disadvantage of this method is that it is possible to introduce air bubbles when changing the direction of flow, so we had to introduce a bubble trap between the syringe pump and the chip to avoid this. Lastly, we varied the flow rate from 1 to 3.6 mL/h. The trapping rates ranged from 42% to 73% and the retention rates from 38% to 54%. We found that a low flow rate can result in the formation of microcolonies, and, although high flow can still result in the loss of some of the trapped cells during operation, overall trapping and retention rates increased with flow. We settled on a maximum flow of 3.6 mL/h because syringes need to be changed manually and higher flows would imply a more frequent change of the syringe and more media use, which was impractical.

### 3.3. Performance Test of the Setup

Using the parameters found in the optimization stage, we tested the chip by measuring cell retention. We grew cells overnight (24 h) in minimal media with 2% Glucose (*w*/*v*) diluted 20 μL in 5 mL of fresh media and grew it for 12 h. This gave us an OD600 of 0.1–0.2. We observed a very clear exponential phase for times between 12 and 17 h. We centrifuged the cells for 5 min at 5000 RPM and re-suspended in 1/5 of the original volume. We loaded the cells, started the flow and measured the percentage of retained cells. We observed retention rates between 25% and 55%. These retention rates compare well with equivalent systems: while in some they are not reported [12,14,20], in others they are below 30% [16,21] and the best ones are between 40% and 60% [21]. If the retention rate is the main consideration, higher flows can be used at the cost of more media use and more frequent syringe changes.

### 3.4. Finite Elements Simulation of the Slipstreaming Effect

To design the device, we modeled the slipstreaming effect by numerically solving the Navier–Stokes equations using the finite element method [27] in COMSOL Multiphysics, using the MEMS and Microfluidics modules for incompressible fluid flow. We modeled the effect of fluid flow for different configurations of the pillars and settled on a symmetric three column design (Figure 1D). Figure 2 shows the simulation for a single channel with the inlet on the left and the outlet on the right. The average inlet velocity was calculated from the syringe pump flow (3.6 mL/h) and the transversal area of the chip, resulting in a velocity of 1.17 m/s. The boundary conditions were set as zero pressure on the outlet and zero velocity on the walls. The fluid properties were set using the density and viscosity of SD medium (density = 1000 kg/m^3^, viscosity = 1 cp). We first modeled the whole channel (Figure 2A) and from that simulation we obtained the boundary conditions and average velocity for each trap. This allowed us to do a second, zoomed-in simulation of each trap, from which we obtained the velocity and pressure profiles (Figure 2B,C). This method allowed us to use a fine mesh for the traps, which would be computationally very costly if used for the whole chip.

The velocity profile shows that the lowest velocity is in the middle of the pillars, but there is also a low velocity zone to the right of the rightmost pillar (Figure 1B). The pressure profile along a line bisecting the trap shows that in the zone where the mother cells are trapped there is a gradient, which results in a force towards the pillar (to the left), and in the area where the daughter cell is born there is a gradient, which results in a force (to the right) that carries the daughter cell away (Figure 2C).

### 3.5. Pressure Affects Cell Physiology

One of the main advantages of our device is the trapping of cells with low mechanical pressure, and we considered this to be an important factor because we found studies that show evidence of mechanical pressure affecting the growth rate [23,24]. We decided to do a quick check of whether gene expression was also affected by the pressure cells experience in a microfluidic device. We used a constitutive gene (LEU) tagged with GFP to measure gene expression in a population. After that, we grew the cells in three different conditions: under no pressure, under low pressure (69 kPa) and under higher pressure (103 kPa). These values were chosen based on the typical values used on the MACS (Microfluidic Activated Cells Screening) device, which is a microfluidic device used for pressing cells mechanically [25,26]. After 4 h (when most cells in the population have already had at least one daughter), we measured again the distribution of gene expression in the population and observed statistically significant changes which confirm that pressure can affect gene expression (Figure 3A).

We performed an ANOVA statistical test for equal means for all of the three different groups. We found *p*-value of less than 10^−4^ when comparing the mean of gene expression under no pressure and at 69 or 103 kPa. This indicates that the means for cells under pressure and under no pressure can be assumed different, whereas the test for cells under 69 kPa and cells under 103 kPa gives a *p*-value of 0.092, indicating these two means are not significantly different. Further experiments are required to determine whether there is a threshold pressure from which gene expression is affected or if it is changed continuously as pressure increases in a way similar to what has been found for growth rates under different pressures [23,24]. In any case, our simple experiment shows that gene expression is affected by mechanical pressure in the range used for cell trapping, which supports the claim that pressure in those experiments may affect the physiology of cells. If cell physiology is affected, studies of the aging process could also be affected.

To estimate the pressure exerted by the other Mother Machine devices, we looked at the deformation of the membrane in published images. We used a simple spring model using a surface modulus of 12 N/m [28]. The model used was
(1)σ=Yε
where σ is the stress, Y is the Young’s Modulus and ε is the deformation.

Figure 3B shows the results of the model applied to the available data. The continuous blue line shows the linear relation between deformation and pressure. The yellow area shows the region in which we observed pressure affects gene expression. The highlighted region starts at 69 kPa, but it is possible that gene expression is affected at even lower values. The red and green points show the average values estimated for two other devices found in the literature [16,21]. The dotted lines show the 95% confidence interval for the deformation measurement and corresponding pressure estimate in those devices. We observed that these devices subject cells to pressures that are high enough to affect gene expression.

### 3.6. Measurement Tests on Chip

To test the usability of our design, we used our chip to measure yeast aging dynamics at the single-cell level. We were able to observe individual mother cells trapped during long periods of time and daughter cells as they grow and detach (Figure 4A–H). To illustrate the trapping mechanism, we measured the distances from pillar to cell, mother cell size and daughter cell position and compared them with the distances and pressures obtained from the Multiphysics simulations (Figure 4I–K). We can observe that, in the region where the mother cells are trapped, the pressure gradient generates a force to the left. In the region where the daughter cell is born and grows, the gradient generates a force to the right.

We measured doubling time distributions (Figure 4L) and RLS (Figure 4M) for BY4741 cells in SD medium and found a mean RLS of 19.2 generations, similar to measurements reported in other publications (Table 1). The 95% confidence interval [29] for the RLS is between 18.43 and 20.54 generations. This value matches the range obtained in other works: other authors [16,17,21] reported average RLS times between 18 and 24 generations and 24.2 and 24.7, respectively. There are differences in the details between experiments, and the measured values of RLS times depend on the cell type and conditions used. Liu [16] used S288C cells, while Jin [17,22] used BY4741 cells—similar to ours—but grown in a different medium (SC).

## 4. Discussion

We present a hydrodynamics based microfluidic “mother machine” device for observing cell dynamics by trapping mother cells and washing away daughter cells. The main advantage of this device is that it does not trap cells by pressing them mechanically as in other devices but uses hydrodynamic flow to trap the cells. In other devices, cells are subjected to external physical pressure of hundreds of kilopascals, whereas in our case the pressure difference that the cell is subjected to is of the order of hundreds of pascals. Our device has the added advantage that it can be fabricated with low-cost methods since it does not require masks with resolution under 5 µm. Other systems require a higher resolution because its physical features are used for the trapping directly, whereas in our case the trapping area is determined by the flow profile. Its only drawback is an imperfect retention rate, but since the device allows for imaging of multiple cells in one field of view this can be compensated by starting with a large number of trapped cells. We compared our device to previous designs and summarized the main features in Table 1.

Long-term expression studies or experiments that measure response to changes in media are hard to do in the traditional setup of microcolonies on slides because neighboring cells change the microenvironment in a matter of minutes, and using micromanipulators to separate the cells is extremely time consuming. As with other Mother Machine systems, the removal of daughter cells in our setup allows for the measurement of long-term behavior of cells in stable media conditions, and using multiple input syringes would allow experiments on the response to changing media in individual cells over time. This device is thus adequate for many types of dynamic measurements, such as aging, replication times, response to media changes, drug screening applications and, in general, any experiment where there is interest in observing populations at the single-cell level for extended periods. Since the cells are trapped within 5 μm of a standard coverslip, all standard microscopy techniques, including fluorescence measurements, can be performed with this setup. As an example of use we demonstrated the capabilities of the device by measuring distributions of replications times and Replicative Life Span curves and obtained similar results to previous studies. Although we demonstrated the use of this device with yeast cells, it should be suitable for other cells with diameters ranging 5–10 µm since the trapping effect does not depend on the cell properties. Adjusting the size of the trapping area is in principle possible, although doing so would require moving back to standard photolithography techniques, negating the low-cost advantage of our design.

By removing the bias from pressure induced changes in the metrics of interest, and reducing the technological requirements for creating the chips, we expect that this design will help established labs make cleaner measurements and resource-limited labs to have access to Mother Machine microfluidic devices, thereby advancing fields such as aging and cell growth.

## Figures and Tables

**Figure 1 micromachines-12-00004-f001:**
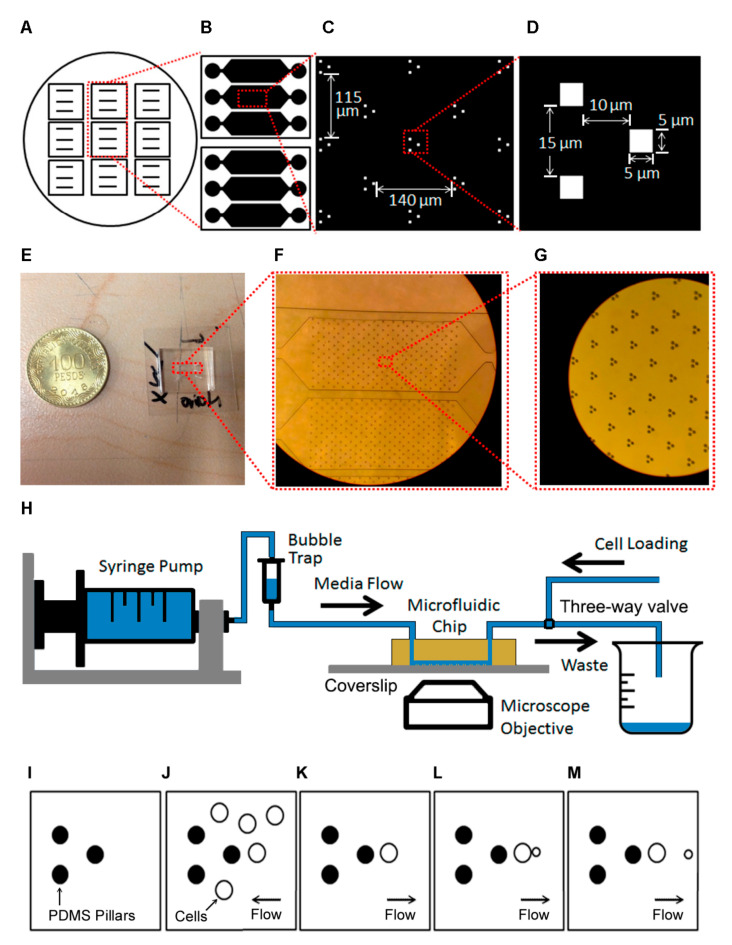
(**A**) The circle represents a silicon wafer with masters for multiple devices. (**B**) Design of each chip, each with three independent channels. (**C**) Individual traps within a channel. We show the vertical and horizontal separations of each of the traps. (**D**) Individual trap dimensions. (**E**). Actual Microfluidic Chip compared with a Colombian Coin. (**F**) Zoom-in of the channels. (**G**) Zoom-in of the pillars. (**H**) Sketch of the experimental setup. (**I**–**M**) Sketch of trap operation. (**I**) Empty trap. (**J**) Concentrated cells are inserted through the outlet. (**K**) Normal flow is started, and only cells that are in the trapping area remain. (**L**) Cells grow and reproduce. The flow directs budding cells to the right. (**M**) When the daughters detach, the flow takes them away.

**Figure 2 micromachines-12-00004-f002:**
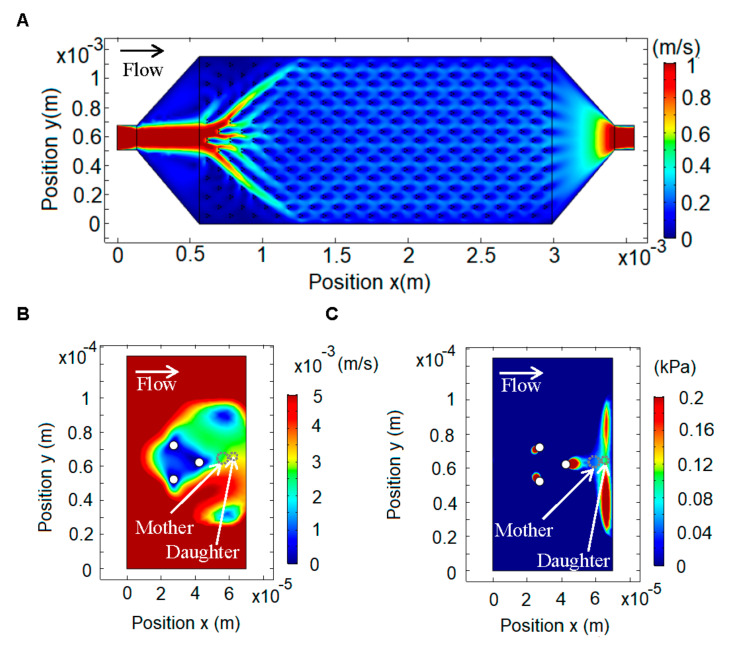
Finite element simulation using COMSOL Multiphysics. (**A**) Velocity profile in a single mother machine channel, with the input on the left and the output on the right, and within it 30 columns, 15 of which have nine rows and 15 of eight rows for a total of 255 traps. (**B**) Flow velocity profile in a single trap. (**C**) Pressure profile in a single trap.

**Figure 3 micromachines-12-00004-f003:**
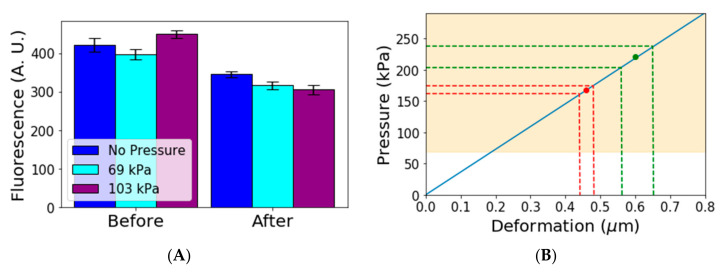
(**A**) Gene expression under different pressure conditions. Bars show average gene expression. Bars on the left show the average gene expression before starting the experiment. Bars on the right show the average gene expression after 4 h for three different pressure conditions. Error bars show 95% confidence intervals (**B**) Blue line shows a simple elastic model relating the deformation to the pressure. The green and red dots represent average deformations and estimated pressure in two other devices [16,21]. The green and red dotted lines show confidence intervals for deformation and pressure. The yellow area represents the region in which we have observed that pressure affects gene expression.

**Figure 4 micromachines-12-00004-f004:**
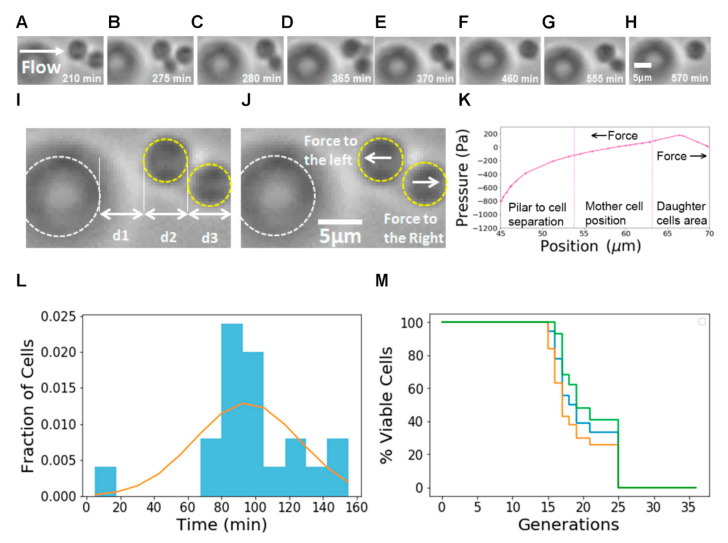
(**A**–**H**) Time-lapse Bright Field images (video frames) showing the dynamics of a single mother cell with two daughters. (**A**) Mother cell with first daughter cell before detachment. (**B**) Mother cell with second daughter cell. (**C**) First daughter is detached, and second daughter continues growing. (**D**) Mother cell with third daughter cell. (**E**) Second daughter cell is detached, and third daughter cell is still growing. (**F**) Third daughter cell is detached. (**G**) Fourth daughter cell growing before detachment. (**H**) Fourth daughter cell is detached. (**I**) Pillar and cell position. PDMS pillars are highlighted in white and cells are highlighted in yellow. (**J**) Sketch of the forces felt by the cells upon detachment. (**K**) Typical pressure profile for a trap. This pressure profile was obtained from Multiphysics simulations. The leftmost section represents the distance from the pillar to the mother cell (d1 in (**I**)). The central section shows the capture area of the mother cell and its approximate size (d2 in (**I**)). The rightmost section shows the region where the daughter cells are born (d3 in (**I**)). Mother cells are trapped due to the action of a force to the left and the daughter cells separate due to a force to the right. (**L**) Measured distribution of replication times with fit to model. (**M**) RLS measurements on our microfluidic chip. The blue line represents the Kaplan–Meier estimator, the green line represents the upper confidence interval limit and the yellow line represents the lower confidence interval limit.

**Table 1 micromachines-12-00004-t001:** Comparison of different Mother Machine Devices.

Year	Name	Mech. Pressing	Requires Under 5 um Resolution	Daughter Removal	Yeast Strain	Medium	Trap Rate	Ret. Rate	<RLS> (gen)
2020	Slipstreaming Mother Machine	No	No	Yes	BY4741	2% Glucose SD	~70%	~50%	19.2
2017, 2019	Cell Traps [17,22]	Yes	Yes	Yes	BY4741	SC + 2% Dextrose	~93%	~75%	18–24
2015	Yeast Replicator [16]	Yes	Yes	Yes	BY Backgr.	min + 2% Glucose	~70%	~50%	24.2
2014	Alcatras 1 [21]	Yes	Yes	Yes	S288C	2% Glucose SC	~90%	~70%	24.7
2014	Alcatras 2 [21]	Yes	Yes	Yes	S288C	2% Glucose SC	~90%	~70%	24.7

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
