# Peer review of "Slipstreaming Mother Machine: A Microfluidic Device for Single-Cell Dynamic Imaging of Yeast"

_micromachines, 2020, doi:10.3390/mi12010004_

Round 1
Reviewer 1 Report
The authors reported a microfluidic device to trap and manipulate cells with low mechanical pressure to minimize its effect on cell physiology. However, the following concerns need to be addressed before publication.
(1) Page 1 line 43 needs references.
(2) Page 3 line 110, this is the second time to mention “Synthetic Dropout (SD) medium” in the manuscript. The authors should explain the full term when they mention it the first time (line 93). Please also check the SC medium.
(3) Line 128, the authors should specify which subfigure they mentioned, not just Figure 1. Please also check other figures.
(4) Line 129, please draw a diagram for the experimental procedure. Explaining only in words is not clear.
(5) Figure 1C and 1D, the dimension labels are not clear. For example, it is hard to tell where are the start and the end of the arrow point to.
(6) Did the authors do mesh independence study in the manuscript? Will the mesh size affect the results?
(7) Figure 3A needs statistical tests.
(8) Line 235, “We assumed a surface modulus of 12 N/m” needs references.
(9) Line 247, what are “two other devices”?
(10) Figure 4A-I need scale bars. Also, please point out which is the pillar and which are cells in these figures.
(11) Figure K, please explain how the pressure is measured?
Author Response
We thank the reviewer for the comments, and apologize profusely for the extra effort they had to do because of our mistake in sending the wrong version (without final edits). The redaction of all sections after the introduction is very different in the final version. Below is our point by point response to the comments.
(1) Page 1 line 43 needs references.
We included the reference to the original paper reporting measurements of RLS (Mortimer and Johnston 1959)
(2) Page 3 line 110, this is the second time to mention “Synthetic Dropout (SD) medium” in the manuscript. The authors should explain the full term when they mention it the first time (line 93). Please also check the SC medium.
We included the composition of SD and SC media in line 105:
Every liter of SD medium contains 6700 mg of YNB, 750 mg of DO supplement, 20 g D-Glucose, 20 mg Uracil and 20 mg Histidine. Every liter of SC medium contained 6700 mg YNB (Yeast Nitrogen Base), 2000 mg DO supplement, 20g D-Glucose, 80 mg D-L Tryptophan, 80 mg D Histidine and 80 mg D-Uracil.
(3) Line 128, the authors should specify which subfigure they mentioned, not just Figure 1. Please also check other figures.
We now specify the subfigure throughout. That specific paragraph is now
We designed and fabricated a novel microfluidic platform (Fig 1 A-H) that can be used to measure single cell dynamics in a population of yeast cells. The microfluidic device at the core of the platform has three separate channels that permit the running of experiments with different media or strains independently (Figure 1B). The device fits in a 22 mm *22 mm coverslip (Figure 1E). Each channel contains an array of 255 trapping units, with 140μm between units in the direction of the flow and 115μm between lines of units (Figure 1C). Each unit consists of three PDMS pillars of 5μm height that extend from the ceiling to the glass floor (Figure 1D). Cells are trapped to the right of the rightmost pillar while the other two pillars are used for flow focusing (Figure 1D).
(4) Line 129, please draw a diagram for the experimental procedure. Explaining only in words is not clear.
We added a sketch of the device operation, and modified the setup sketch to make it clearer. We added the following experimental procedure description and description of the sketch of operation.
The system setup is sketched in Figure 1H. First, the chip is set on the (inverted) microscope and fresh media is loaded into a syringe pump, which is connected to a bubble trap and that in turn is connected to the chip inlet. The outlet is connected to a three way valve, with exits connecting to the waste beaker and a tube used for initial loading of the cells. The cells are then loaded manually through that tube, making sure they don’t reach the bubble trap. The three way valve is then set to the waste beaker and flow from the syringe is started. The operation of the trap during this process is sketched in Figures 1I-M.
I-M Sketch of trap operation. I. Empty trap. J. Concentrated cells are inserted through the outlet. K. Normal flow is started, and only cells that are in the trapping area remain. L. Cells grow and reproduce. The flow directs budding cells to the right. M. When the daughters detach, the flow takes them away.
(5) Figure 1C and 1D, the dimension labels are not clear. For example, it is hard to tell where are the start and the end of the arrow point to.
We have adjusted and enlarged the end bars to make the dimensions easier to read.
(6) Did the authors do mesh independence study in the manuscript? Will the mesh size affect the results?
We did not do a full mesh independence trial, but we did try various mesh sizes to improve resolution around the traps. We included text indicating this:
We first modeled the whole channel (Figure 2A) and from that simulation we obtained boundary conditions and average velocity for each trap. This allowed us to do a second, zoomed-in simulation of each trap, from which we obtained the velocity and pressure profiles (Figure 2B and 2C). This method allowed us to use a fine mesh for the traps, which would be computationally very costly if used for the whole chip.
(7) Figure 3A needs statistical tests.
We included the following paragraph discussing the statistical significance of the experiments in Figure 3A:
We performed an ANOVA statistical test for equal means for all of the three different groups. We found p-value of less than 10-4 when comparing the mean of gene expression under no pressure and at 69 kPa or 103 kPa. This indicates that the means for cells under pressure and under no pressure can be assumed different, whereas the test for cells under 69kPa and cells under 103kPa gives a p-value of 0.092, indicating these two means are not significantly different. Further experiments are required to determine whether there is a threshold pressure from which gene expression is affected or if it is changed continuously as pressure increases in a way similar to what has been found for growth rates under different pressures [12].
(8) Line 235, “We assumed a surface modulus of 12 N/m” needs references.
We included the reference (Smith E.A. et al., 2000):
We used a simple spring model using a surface modulus of 12 N/m [17].
(9) Line 247, what are “two other devices”?
We included the references to the other two devices (Swain’s and Acar’s):
The red and green points show the average values estimated for two other devices found in literature [10, 14].
(10) Figure 4A-I need scale bars. Also, please point out which is the pillar and which are cells in these figures.
We added a scale bar to figure 4H, which is the same for all other frames, and highlighted the pillars in white and the cells in yellow in figures 4I and 4J.
(11) Figure 4K, please explain how the pressure is measured?
The pressure profile in figure 4K was obtained from the simulations. We now clarify this both in the paragraph
To illustrate the trapping mechanism we measured the distances from pillar to cell, mother cell size and daughter cell position and compared them with the distances and pressures obtained from the Multiphysics simulations (Figure 4 I-K).
and in the figure legend
K. Typical pressure profile for a trap. This pressure profile was obtained from Multiphysics simulation

Reviewer 2 Report
Summary: This manuscript by Durán et al. describes the development of a microfluidic device for the trapping of individual yeast cells in a mechanically-free manner for aging studies. The trapping of cells is based on using microstructured posts inside the microfluidic chip for the generation of a pressure gradient. Mother cells are trapped in this region and the daughter cells, once detached, are forced to flow away with the fluid flow. This is a simple but potentially interesting approach. However, in the opinion of this referee, this manuscript needs to be significantly improved before being considered for publication. It lacks a lot of important information, clarifications, and critical discussion. My main comments are summarized below.
General Comments:
1) Language: while the Introduction section is very clear and well-written, the rest of the manuscript needs to be significantly polished. A. The manuscript should be read (and corrected) by a native English speaker.
Minor points:
- Abstract: Remove the acronym RLS from the abstract since it is not employed.
- Page 2, lines 86: Replace “PDMS” by “pre-polymer”.
- Page 2, lines 86 (and all throughout the manuscript): use alwaus ºC instead of “degrees (Celsius)”. It is mixed in the manuscript without any rational.
- Page 2, line 87: Replace “holes” by “inlet/outlets”.
- Page 3, line 93: Define “SD” and “SC” in this section.
- Page 3, line 105: Micromanager is written with “M” (capital letter).
- Page 3, lines 118: Replace “mm square” by “mm^2”.
- Page 3, lines 120: Replace “microns” by “um”.
- Figure 1: Authors may consider adding a scheme describing the working procedure of the cell trapping for clarity.
Major issues:
- Pages 2-3, Materials and Methods: This section must be significantly improved. It lacks a lot of fundamental information. For instance, how was the chip designed (e.g., AutoCAD,...)? Did the authors employ a photomask? If so, was it made of acetate or chromium? Who is the supplier of all the materials/equipment employed? Which liquid was employed to sonicate the chips? How long was the O2 plasma used? Are the objectives air or immesion? Which is their NA?
- Section 3.1: the dimensions of the chip should be included in the M&M section and not in the Results.
- Page 3, lines 127: How many cells were loaded in the device? Were they loaded manually or using a (syringe) pump?
- Figure 2: Why the simulation profile seems to be asymmetric (Fig 2A and 2B)? This must be corrected/clarified. In addition, are the units (m/s) the velocity legend correct? It seems to be a ^-3 missing. Finally, I´d suggest to replace the bold spots in Figure 2C by dashed transparent circumferences to better see the simulation image.
- Figure 3: Add statistical significance analysis.
- Figure 4: The images are of low resolution. Please, provide clearer images.
- Section 3.6: This sub-section should be included inside the Discussion section.
- Did the authors evaluate the performance of the device with other biological entities (other yeast, mammalian cells,..etc? Which optimizations may need to be performed for this objective? This should be discussed.
- Did the authors investigate the use of the device for the addition of drugs (e.g., cytoskeleton) and analyze whether it is compatible for drug screening applications? This is an important point that the authors must address.
- Is the system compatible with fluorescence microscopy? This might be important for the on-chip study of gene expression.
- The Discussion section must be improved taking into consideration the above-mentioned points.
Author Response
We thank the reviewer for the comments, which should greatly improve the paper, and apologize profusely for the extra effort they had to do because of our mistake in sending the wrong version (without final edits). The redaction of all sections after the introduction is very different in the final version. Below is our point by point response to the comments.
General Comments:
1) Language: while the Introduction section is very clear and well-written, the rest of the manuscript needs to be significantly polished. A. The manuscript should be read (and corrected) by a native English speaker.
The reviewer is absolutely right and we apologize again for the inconvenience. The redaction of most sections had multiple changes in the final edit version and has been edited again to incorporate the comments and suggestions of the reviewers.
Minor points:
- Abstract: Remove the acronym RLS from the abstract since it is not employed.
We removed the acronym RLS from the abstract.
- Page 2, lines 86: Replace “PDMS” by “pre-polymer”.
We rewrote the line as
Curing agent was mixed with pre – polymer for PDMS
- Page 2, lines 86 (and all throughout the manuscript): use alwaus ºC instead of “degrees (Celsius)”. It is mixed in the manuscript without any rational.
We replaced “degrees Celsius” with °C throughout.
- Page 2, line 87: Replace “holes” by “inlet/outlets”.
We changed it to
… and inlet/outlet channels were punched ...
- Page 3, line 93: Define “SD” and “SC” in this section.
We included the composition of SD and SC media in line 105:
Every liter of SD medium contains 6700 mg of YNB, 750 mg of DO supplement, 20 g D-Glucose, 20 mg Uracil and 20 mg Histidine. Every liter of SC medium contained 6700 mg YNB (Yeast Nitrogen Base), 2000 mg DO supplement, 20g D-Glucose, 80 mg D-L Tryptophan, 80 mg D Histidine and 80 mg D-Uracil.
- Page 3, line 105: Micromanager is written with “M” (capital letter).
It has been corrected.
- Page 3, lines 118: Replace “mm square” by “mm^2”.
We replaced the line by:
The device fits on a 22 mm *22 mm coverslip (Figure 1E).
- Page 3, lines 120: Replace “microns” by “um”.
It has been corrected.
- Figure 1: Authors may consider adding a scheme describing the working procedure of the cell trapping for clarity.
We included this sketch of the operation of the trap.
I-M Sketch of trap operation. I. Empty trap. J. Concentrated cells are inserted through the outlet. K. Normal flow is started, and only cells that are in the trapping area remain. L. Cells grow and reproduce. The flow directs budding cells to the right. M. When the daughters detach, the flow takes them away.
Major issues:
- Pages 2-3, Materials and Methods: This section must be significantly improved. It lacks a lot of fundamental information. For instance, how was the chip designed (e.g., AutoCAD,...)? Did the authors employ a photomask? If so, was it made of acetate or chromium? Who is the supplier of all the materials/equipment employed? Which liquid was employed to sonicate the chips? How long was the O2 plasma used? Are the objectives air or immesion? Which is their NA?
We expanded the M&M section. It is now:
2. Materials and Methods
Master Design and Fabrication
We used an SF100 (ScoTech) Micropatterning UV projector instead of a high resolution printed mask. This projector requires a virtual mask in bmp (bitmap) format, so we designed it pixel by pixel using Illustrator and Paint. One pixel in the design is equivalent to a 5µm by 5µm square in the physical chip. In a cleanroom, we spun SC 1827 Photoresist at 2000 RPM for 1 minute and then soft baked it at 115°C for 50 s. We then exposed it for 50s to UV light using the projector and developed the master for 1 minute before hard baking it at 120 °C for 1 min.
Polydimethylsiloxane (PDMS) Chips Fabrication
Curing agent was mixed with pre – polymer for PDMS in a proportion of 1:10 and cured in an oven at 65°C. The PDMS was then peeled off the master and inlet/outlet channels were punched with a 0.75 mm biopsy punch. Chips were then cut and sonicated in isopropanol for 30-40 min, then blown, dried and heated at 65°C for 4 h. Glass coverslips were cleaned in 1M KOH for 30 min, then sonicated in milliQ water for 10 min. After that, the coverslips were blow-dried and heated at 65°C for 10 min. The PDMS chips were cleaned with magic tape and treated with oxygen plasma for 1 min for bonding and hot bonded for 1 min at 150°C.
Chip Dimensions
The device fits on a 22 mm * 22 mm coverslip. Each channel contains an array of 255 trapping units, with 140μm between units in the direction of the flow and 115 μm between lines of units. Each unit consists of three PDMS pillars of 5 μm height that extend from the ceiling to the glass floor.
Media Preparation
We grew the cells in either Synthetic Dropout (SD) or Synthetic Complete (SC) medium. Every liter of SD medium contains 6700 mg of YNB, 750 mg of DO supplement, 20 g D-Glucose, 20 mg Uracil and 20 mg Histidine. Every liter of SC medium contained 6700 mg YNB (Yeast Nitrogen Base), 2000 mg DO supplement, 20g D-Glucose, 80 mg D-L Tryptophan, 80 mg D Histidine and 80 mg D-Uracil.
Cell Growth and Insertion in Device
We recovered cells from -80°C storage and grew them in SD Medium (2% Glucose) for 24 hours. We diluted 20 ml in 5 ml fresh medium and grew the cells at 30°C and 245 RPM for 12 hours to get them back to an exponential phase (0.05-0.1 OD600). We then took media with these cells and centrifuged it at 5000 RPM for 5 minutes and then resuspended in 1/5 of the original volume. The cells were then manually inserted into the chip in the direction opposite to the normal flow before turning on the syringe pump and starting the experiment.
Data acquisition and analysis
Growth experiments were performed on a Zeiss inverted microscope controlled with Micromanager. We used 40X (air), 60X (air) and 100X (immersion) objectives. Images were acquired using a Photometrics CoolSnap Camera EZ CCD Camera with 1392 x 1040. Media was delivered using a syringe pump (Harvard Apparatus) at a rate of 3.6 milliliters/h. Expression experiments were performed on a Nikon Ti Eclipse inverted microscope with a 100X (immersion) objective.
Growth conditions for measuring genetic expression under pressure
Cells were grown in SD medium supplemented with aminoacids (Uracil and Histidine) and Glucose. Cells were grown overnight (24 h) from a plate. We took 20 ml from the overnight culture and diluted it in 10 ml fresh media, then let it grow for 12 h.
- Section 3.1: the dimensions of the chip should be included in the M&M section and not in the Results.
Agreed, see above.
- Page 3, lines 127: How many cells were loaded in the device? Were they loaded manually or using a (syringe) pump?
We were able to load an average of 179.5 cells per channel. This is now stated in line 157:
This resulted in a trapping rate of approximately 70%, or about 180 cells per channel.
The loading was done manually. This is now stated within the following paragraph:
The system setup is sketched in Figure 1H. First, the chip is set on the (inverted) microscope and fresh media is loaded into a syringe pump, which is connected to a bubble trap which in turn is connected to the chip inlet. The outlet is connected to a three way valve, with exits connecting to the waste beaker and a tube used for initial loading of the cells. The cells are then loaded manually through that tube, making sure they don’t reach the bubble trap. The three way valve is then set to the waste beaker and flow from the syringe is started. The operation of the trap during this process is sketched in Figures 1I-M.
- Figure 2: Why the simulation profile seems to be asymmetric (Fig 2A and 2B)? This must be corrected/clarified. In addition, are the units (m/s) the velocity legend correct? It seems to be a ^-3 missing. Finally, I´d suggest to replace the bold spots in Figure 2C by dashed transparent circumferences to better see the simulation image.
We changed Figures 2B and 2C as suggested. The reason for the small asymmetry is that no traps are exactly in the middle of the main channel, so the boundary conditions are not exactly the same on the top and bottom of the section that was simulated with the fine mesh. The exact values depend on which trap we focus on. This should be clearer from the following paragraph:
We first modeled the whole channel (Figure 2A) and from that simulation we obtained boundary conditions and average velocity for each trap. This allowed us to do a second, zoomed-in simulation of each trap, from which we obtained the velocity and pressure profiles (Figure 2B and 2C). This method allowed us to use a fine mesh for the traps, which would be computationally very costly if used for the whole chip.
- Figure 3: Add statistical significance analysis.
We included the following paragraph discussing the statistical significance of the experiments shown in Figure 3A:
We performed an ANOVA statistical test for equal means for all of the three different groups. We found p-value of less than 10-4 when comparing the mean of gene expression under no pressure and at 69 kPa or 103 kPa. This indicates that the means for cells under pressure and under no pressure can be assumed different, whereas the test for cells under 69kPa and cells under 103kPa gives a p-value of 0.092, indicating these two means are not significantly different. Further experiments are required to determine whether there is a threshold pressure from which gene expression is affected or if it is changed continuously as pressure increases in a way similar to what has been found for growth rates under different pressures [12].
- Figure 4: The images are of low resolution. Please, provide clearer images.
The images provided are zoomed in sections of a time series; we unfortunately do not have a way to increase the resolution and we cannot currently redo the experiments due to covid-related restrictions in our university. To help with the interpretation we highlighted the pillars in white and the cells in yellow in figures 4I and J.
- Section 3.6: This sub-section should be included inside the Discussion section.
It has been moved to the discussion section.
- Did the authors evaluate the performance of the device with other biological entities (other yeast, mammalian cells,..etc? Which optimizations may need to be performed for this objective? This should be discussed.
We did not evaluate other types of cells, but since the hydrodynamic effect we use does not strongly depend on the details of the cell beyond its dimensions, it should work for cells of the same size range. While the idea could in principle be extended to other sizes, we do not explore it because varying the size of the pillars would require a higher resolution than the UV projector can provide. This would require using standard high-resolution photolithography techniques, and we would lose the low-cost aspect of our design. Working in a low-resource country, this advantage is very important to us as we hope our design will allow other low-resource labs to start working with these techniques. We added a brief discussion of this point:
Although we demonstrated the use of this device with yeast cells, it should be suitable for other cells with sizes with a diameter range of 5-10 micrometers since the trapping effect does not depend on the cell properties. Adjusting the size of the trapping area is in principle possible, although doing so would require moving back to standard photolithography techniques, negating the low cost advantage of our design.
- Did the authors investigate the use of the device for the addition of drugs (e.g., cytoskeleton) and analyze whether it is compatible for drug screening applications? This is an important point that the authors must address.
We did not investigate the drug screening applications, but since it shares with other mother machine designs the characteristic of the cell being in contact with flowing media, it is very well suited for drug screening or changing-media experiments. The interplay of changes in the cytoskeleton and membrane and the expression changes due to pressure is an area where we are very interested in pursuing, but for this paper we wanted to focus on the device itself. We added the following comment to the discussion
As with other Mother Machine systems, the removal of daughter cells in our setup allows for the measurement of long term behavior of cells in stable media conditions, and using multiple input syringes would allow experiments on the response to changing media in individual cells over time. This device is thus adequate for many types of dynamic measurements, such as aging, replication times, response to media changes, drug screening applications and, in general, any experiment where there is interest in observing populations at the single-cell level for extended periods.
- Is the system compatible with fluorescence microscopy? This might be important for the on-chip study of gene expression.
It is, and we added a sentence to highlight this:
Since the cells are trapped within 5 mm of a standard coverslip, all standard microscopy techniques, including fluorescence measurements, can be performed with this setup.

Round 2
Reviewer 2 Report
The authors satisfactorily addressed most of my comments improving the quality of the manuscript. However, there are still some innacuracies that must be polished before accepting the manuscript. Some examples include:
- 5µm by 5µm square, in Materials and Methods -> the word "square" should be deleted.
- In Materials and Methods -> Replace "milliliters/h" by "mL/h"
- Figure 1: Some of the figures panels are not announced in order in the main text. Some others are not announced at all. This must be corrected.
I stronhly recommend the authors to carefully check all the manuscript and correct this type of mistakes.
After the authors address all these issues, the manuscript can be accepted for publication.
Author Response
We thank the reviewer for the positive assessment and the detail of his observations. We have done all suggested edits.
- 5µm by 5µm square, in Materials and Methods -> the word "square" should be deleted.
It has been deleted.
- In Materials and Methods -> Replace "milliliters/h" by "mL/h"
It has been replaced.
- Figure 1: Some of the figures panels are not announced in order in the main text. Some others are not announced at all. This must be corrected.
We have changed the order of the sentences to match the order in the figure and added two referencing Figures 1A, F and G:
A silicon wafer contains masters for 9 devices (Figure 1A).
Figures 1F and 1G show a zoom-in of the channel and traps.
We have reviewed the whole text again and made a few minor corrections.
